# Probability of Deriving a Yearly Transition Probability Matrix for Land-Use Dynamics

**Shigeaki F. Hasegawa [1],\*** and **Takenori Takada [2]**

1   Institute of Low Temperature Science, Hokkaido University, N19W8, Kita-Ku, Sapporo 060-0819, Japan
2   Faculty of Environmental Earth Science, Hokkaido University, N10W5, Kita-Ku, Sapporo 060-0810, Japan; takada@ees.hokudai.ac.jp
\*   Correspondence: shasegaw@lowtem.hokudai.ac.jp; Tel.: +81-11-706-7660

**Abstract:** Takada's group developed a method for estimating the yearly transition matrix by calculating the *m*th power roots of a transition matrix with an interval of m years. However, the probability of obtaining a yearly transition matrix with real and positive elements is unknown. In this study, empirical verification based on transition matrices from previous land-use studies and Monte-Carlo simulations were conducted to estimate the probability of obtaining an appropriate yearly transition probability matrix. In 62 transition probability matrices of previous land-use studies, 54 (87%) could provide a positive or small-negative solution. For randomly generated matrices with differing sizes or power roots, the probability of obtaining a positive or small-negative solution is low. However, the probability is relatively large for matrices with large diagonal elements, exceeding 90% in most cases. These results indicate that Takada et al.'s method is a powerful tool for analyzing land-use dynamics.

**Keywords:** land-use dynamics; transition probability matrix; yearly transition; power root of matrix

## 1. Introduction

Land-use and land-cover change with natural processes and human activities, which further depends on ecological, economic, political institutional, and social constraints [1]. Thus, studying land-use/cover change (LUCC) may contribute to better understanding of the interaction between environmental and human-driven processes and finding key processes within the local human–environment system [2]. Several approaches were developed to understand, analyze and evaluate LUCC [3–6]. Among them, the probability-based transition matrix approach has been used to analyze, compare, and predict LUCC over specific periods with a stationary Markov model [7–10]. In this approach, two maps of a single site for two points in time are classified into the same set of land-use/cover categories and the transition probabilities between the categories are estimated by comparing these two maps [11]. The transition probability matrix, $T$, whose interval is *m* years, is used to calculate the projection of the area of LUCC, $x_{t+m}$ as

$$x_{t+m} = x_t \cdot T, \tag{1}$$

where $x_t$ is a row vector representing the proportion of each category in time *t*.

The transition probability matrix is useful not only for extracting factors that lead to differences in one time period at a site, but also for comparing the difference of land-use change among several time periods [8]. Transition probability matrices are sometimes obtained by comparing aerophotographs or satellite images of the target location, but in some cases, the intervals of the aerophotographs or satellite images differ. For example, consider three aerophotographs of the same place taken in 2000, 2007 and

2012. The LUCC transition probability matrix between 2000 and 2007 can be calculated by comparing 2000 and 2007 aerophotographs. In addition, the transition probability matrix between 2007 and 2012 can be calculated. These two transition probability matrices cannot be compared, because the former matrix reflects LUCC in 7 years, whereas the latter matrix reflects LUCC in 5 years. Such mismatch of aerophotograph or satellite image shooting interval might arise from unsystematic planning of shooting interval. The same situation sometimes occurs for comparing LUCC in several sites, because aerophotographs with the same interval are not always available.

To resolve this problem, the matrices should be adjusted such that they have the same interval. Takada et al. [12] developed a method for estimating the yearly transition matrix by calculating the power root of a transition probability matrix with any interval of the research period. Hereafter, Takada et al. [12]'s annualizing method is referred as TAM for short. TAM has been used by many LUCC researchers [13–17], etc. These studies dealt with various topics, such as agricultural land use, forest management, climate change, deforestation, and urbanization, suggesting that TAM is useful for obtaining the yearly transition matrix in LUCC analysis.

Theoretically, the number of solutions induced from $m$th power roots of an $n \times n$ matrix $T$ is $m^n$, because they are calculated as

$$T^{\frac{1}{m}} = U \begin{pmatrix} (\lambda_1)^{\frac{1}{m}} & & 0 \\ & \ddots & \\ 0 & & (\lambda_n)^{\frac{1}{m}} \end{pmatrix} U^{-1}$$

$$U = (u_1 \cdots u_n),$$

(2)

where $\lambda_i$ is the $i$-th eigenvalue of matrix $T$ and $u_i$ is its corresponding eigenvector. The number of $m$th power root of $\lambda_i$ is generally $m$ for each $i$ and the total number of the combinations is $m^n$ [12]. The $m^n$ solutions include matrices with complex numbers or negative numbers. They are unsuitable for land-use dynamics analyses, such as scenario-based simulation and future prediction. This problem has been discussed in several studies [6,18–21]. However, the possibility of obtaining suitable yearly transition probability matrices with TAM remains unclear.

The aims of this study are to (1) clarify the possibility of obtaining yearly transition probability matrices with real field data set, (2) clarify the theoretical possibility of obtaining yearly transition probability matrices, and (3) explain the difference between real and theoretical results and examine the validity of TAM. In this study, we estimated the probability of acquiring a positive or small-negative solution via TAM. Empirical verification was conducted with 62 transition matrices obtained from previous land-use change studies. Monte-Carlo simulations were conducted with randomly generated matrices and biased matrices to estimate the probability of acquiring suitable solutions. Furthermore, we discuss the effectiveness of TAM.

## 2. Materials and Methods

### 2.1. Empirical Verification

In this study, the possibility of obtaining yearly transition probability matrices with actual transition probability matrices was examined. From 34 previous studies on land-use change (Table 1), 62 transition probability matrices were obtained.

**Table 1.** Data sources for empirical verification.

| Data Source | Country | Geographical Place |
|---|---|---|
| Barima et al. 2010 [22] | Ivory Coast | Tanda |
| Bogaert et al. 2011 [23] | Benin | Banokoara |
| Chust et al. 1999 [24] | Spain | Minorca |
| Deng et al. 2009 [25] | China | Hangzhou, Zhejiang province |
| Ediger & Huafang 2006 [26] | China | Western Yunnan |
| Ferreira Filho & Horridge 2014 [27] | Brazil | Sao Paulo, Mato Grosso and nationwide |
| Flamenco-Sandoval et al. 2007 [28] | Mexico | Chiapas state |
| Freitas et al. 2010 [29] | Brazil | Sao Paulo state |
| Guan et al. 2011 [30] | Japan | Saga |
| Günlü et al. 2009 [31] | Turkey | Rize |
| Hall et al. 1991 [32] | USA | Minnesota |
| Hu et al. 2013 [33] | China | Fuzhou City |
| Jasinski et al. 2005 [34] | Brazil | Mato Grosso |
| Jia et al. 2004 [35] | China | Xinjian |
| Kane et al. 2014 [36] | USA | Phoenix, Arizona |
| LaGro Jr. & DeGloria 1992 [37] | USA | New York State |
| Mas et al. 2004 [38] | Mexico | nationwide |
| Matsushita et al. 2006 [39] | Japan | Lake Kasumigaura basin |
| Mendoza et al. 2011 [40] | Mexico | Lake Cuitzeo Watershed |
| Parès-Ramos et al. 2008 [41] | Puerto Rico | nationwide |
| Peña et al. 2007 [42] | Spain | Marina Baixa catchment |
| Pueyo & Alados 2007 [43] | Spain | Middle Ebro Valley |
| Rutherford et al. 2008 [44] | Switzerland | nationwide |
| Silva et al. 2011 [45] | Portugal | Agueda, Macao and Braganca |
| Solon 2009 [46] | Poland | Warsaw metropolitan area |
| Takada et al. 2010 [12] | Japan | Abukuma |
| Thomlinson et al. 1996 [47] | Puerto Rico | Luquillo |
| Weng 2001 [48] | China | Zhujiang Delta |
| Yu & Ng 2006 [49] | China | Panyu, Guandzhou |
| Yuechen 2008 [50] | China | 13 provinces in Northern China |
| Zarin et al. 2001 [51] | Brazil | Amapa state |

These matrices were annualized with the software developed by Takada et al. [12] whose name was "annualmatrix.exe" in the following URL, https://taktakada.github.io/esoftdownload.html, to determine the number of positive and small-negative solutions. Among 62 transition probability matrices used in the empirical verification, 48 matrices were supplied together with the initial and final area size (or proportion) of each category (the row of "Area data" in Table 2). For these matrices, the row vector of the final area size, $v_{fin}$ can be calculated as

$$v_{fin} = v_{init} \cdot T, \tag{3}$$

where $T$ is the transition probability matrix and $v_{init}$ is the row vector of the initial area size. Assume that $T$ is a transition probability matrix in $m$ years and $A$ is a $m^{th}$ power root of the matrix calculated by TAM. The row vector of the estimated final area size, $v_{est}$ would be calculated as

$$v_{est} = v_{init} A^m. \tag{4}$$

Errors in the estimation of the annualization are calculated as the sum of differences in each category between the real and estimated area sizes, using the following formula,

$$\sum \frac{|v_{est} - v_{fin}|}{v_{fin}}. \tag{5}$$

The Pearson's correlation coefficient between the estimated error and study area was calculated using statistical software R version 2.11.1 [52]. The Kendall rank correlation coefficient between the estimated error and number of classes, and between the estimated error and study period was also calculated using R version 2.11.1 as well as Kendall library [53].

## *2.2. Monte-Carlo Simulation*

To estimate the probability of obtaining a positive solution of the power root matrix, we conducted Monte-Carlo simulations using a randomly generated matrix. LUCC transition probability matrices have the characteristic of all the row sums being always 1.0. Assume that $P$ is a $n \times n$ transition probability matrix and $p_{ij}$ is an element of the $P$. The row sum

$$\sum_{i=1}^{n} p_{ij} = 1, \tag{6}$$

for all $j$. In most LUCC studies, the total area or number of grids of the target area neither increases nor decreases during the study period. Thus, randomly generated matrices should meet this requirement. In this study, an $n \times n$ random matrix was generated based on the "broken stick" method proposed by Takada et al. [54]. First, $n - 1$ random numbers were generated from the uniform distribution ranging from 0 to 1 using R version 2.11.1 [52]. and then sorted in the ascending order. A line (stick) with length 1 is broken into $n$ pieces using the random numbers as breaking points. The lengths of broken lines are used as $n$ random numbers, whose sum is equal to 1. These random numbers are combined to generate a row vector of $n$ size, and its sum is equal to 1. For example, three random vectors whose sizes were 3 and sum of the elements was 1.0, were generated as

$$
\begin{aligned}
v_1 &= [0.1, 0.4, 0.5], \\
v_2 &= [0.4, 0.3, 0.3], \\
v_3 &= [0.2, 0.2, 0.6].
\end{aligned}
\tag{7}
$$

This procedure was repeated $n$ times and $n$ row vectors were obtained. They were concatenated to form an $n \times n$ matrix $T$ as

$$
T = \begin{bmatrix} 0.1 & 0.4 & 0.5 \\ 0.4 & 0.3 & 0.3 \\ 0.2 & 0.2 & 0.6 \end{bmatrix}.
\tag{8}
$$

The obtained random matrices were annualized with the software developed by Takada et al. [12] to determine the number of positive and small-negative solutions. Monte-Carlo simulations were conducted for matrix sizes ranging from 2 to 9 and power roots of 3, 4, 5, 7, 10, 13, 20, or 30, except for a matrix size of 8 and power root of 20 or 30 and matrix size of 9 and power root of 20 or 30. The simulation was repeated 1000 times in most cases. The procedure was repeated 100 times for matrices with large sizes and power roots because the simulations were time-consuming. The probability of obtaining a positive or small-negative solution was estimated by dividing the number of trials that yielded a positive or small-negative solution by the total number of trials.

## *2.3. Biased Monte-Carlo Simulation*

A fully random matrix was generated using the "broken stick" method. However, the transition probability matrix analyzed for land-use change tends to differ from a random matrix. In many cases, the diagonal elements of a transition matrix are relatively larger than non-diagonal elements (e.g., [8,28,30,40,43,45,55–58]). This may be attributed to the generally constant land-use patterns during the study period or the tendency of self-replacement probability to be high. To simulate a transition probability matrix in land-use dynamics, we generated a series of biased random matrices

(Hereafter, this type of matrix is referred to as a "modified random matrix"). Therefore, the "broken stick" method was not adopted to generate modified random matrices,

The procedure is as follows. First, $n$ random numbers were generated from the F distribution with 1.0 and 0.0 degrees of freedom using R version 2.11.1 [52]. F distribution was used to simulate a skewed distribution, in which the majorities of random numbers were relatively small, while the minority of random numbers were relatively large. Then, the random numbers were combined to produce a row vector $v$, which was corrected such that the sum of the row vector was one as

$$v'_i = v_i - \left(\sum v - 1\right)\frac{v_i}{\sum v}, \tag{9}$$

where $v'$ is a corrected vector. This procedure was repeated $n$ times. For each row vector that constituted a random matrix, the largest element in the row vector was swapped with the element in the diagonal position of the matrix. For example, three random vectors with a size of 3 and sum of elements of 1.0 were generated as

$$\begin{aligned} v_1 &= [0.1, 0.3, 0.6], \\ v_2 &= [0.5, 0.2, 0.3], \\ v_3 &= [0.1, 0.2, 0.7]. \end{aligned} \tag{10}$$

These three vectors, $v_1, v_2, v_3$ were combined as an $3 \times 3$ matrix $T$

$$T = \begin{bmatrix} 0.6 & 0.3 & 0.1 \\ 0.2 & 0.5 & 0.3 \\ 0.1 & 0.2 & 0.7 \end{bmatrix}, \tag{11}$$

by swapping the first and third elements of $v_1$, and by swapping the first and the second elements of $v_2$.

The same procedure was applied for the modified random matrices to obtain the probability of obtaining a positive or small-negative solution by TAM.

## 3. Results

The size (number of categories) of the transition probability matrices ranged from 4 to 10 and the study period ranged from 3 to 52 years among 62 transition probability matrices used in empirical verification. A positive or small-negative solution was obtained from 54 of 62 matrices(87%) (Table 2).

Error estimation was conducted for 42 transition probability matrices, which were supplied together with the initial and final area size of each category and an annual transition probability matrix could be obtained. Estimated errors in transition probability matrices were smaller than 0.05, except for data from Lopéz et al. [59] (No. 25 in Table 2) with 0.081 (Table 2). The Pearson's correlation coefficient between the estimated error and the study area was not significant (r = −0.16, p = 0.30). In addition, the Kendall's $\tau$ between the estimated error and the number of classes was not significant ($\tau$ = 0.213, p = 0.07). Nevertheless, the estimated error and study period were significantly correlated ($\tau$ = −0.31, p = 0.005).

The probability of obtaining a yearly transition matrix was estimated through Monte-Carlo simulations using random matrices (Table 3).

The probability of obtaining a positive solution was very low for different matrix sizes and power roots (Table 3), and it was almost zero for a matrix size greater than 4. The probability of obtaining a positive solution did not increase linearly with the power root. Although the probability of a small-negative solution with a random matrix was higher than that for a positive solution, it was still less than 30% (Table 3). The probability of obtaining a small-negative solution increased with the power root for matrix sizes larger than 4. However, the relationship between a small-negative solution and the power root was not linear for matrix sizes of 2 or 3.

**Table 2.** Empirical verification of obtainingpositive and small-negative solutions with transition probability matrices from previous field studies.

| No. | Data Source | Study Area (km$^2$) | No. of Classes | First year | Study Periods (year) | Area Data | No. of Results Positive | No. of Results Small neg. | Errors |
|---|---|---|---|---|---|---|---|---|---|
| 1 | Bogaert et al. 2011 [23] | 192 | 4 | 1972 | 34 | with | 0 | 1 | 0.043 |
| 2 | Chust et al. 1999 [24] | 700 | 5 | 1984 | 8 | with | 0 | 1 | 0.0021 |
| 3 | Deng et al. 2009 [25] | 720 | 5 | 1996 | 10 | with | 1 | 0 | $1.1 \times 10^{-6}$ |
| 4 | Deng et al. 2009 [25] | 720 | 5 | 1996 | 4 | with | 0 | 1 | 0.00080 |
| 5 | Deng et al. 2009 [25] | 720 | 5 | 2000 | 3 | with | 1 | 0 | $5.6 \times 10^{-7}$ |
| 6 | Deng et al. 2009 [25] | 720 | 5 | 2003 | 3 | with | 1 | 0 | $1.1 \times 10^{-6}$ |
| 7 | Ediger & Huafang 2006 [26] | 42 | 4 | 1989 | 12 | with | 0 | 1 | 0.020 |
| 8 | Ferreira Filho & Horridge 2014 [27] | 248,000 | 4 | 1995 | 11 | with | 0 | 1 | 0.00075 |
| 9 | Ferreira Filho & Horridge 2014 [27] | 903,000 | 4 | 1995 | 11 | with | 0 | 0 | – |
| 10 | Ferreira Filho & Horridge 2014 [27] | 8,515,000 | 4 | 1995 | 11 | with | 0 | 1 | $1.0 \times 10^{-5}$ |
| 11 | Flamenco-Sandoval et al. 2007 [28] | 5755 | 7 | 1986 | 9 | with | 0 | 1 | 0.0020 |
| 12 | Flamenco-Sandoval et al. 2007 [28] | 5755 | 7 | 1995 | 5 | with | 0 | 1 | 0.0063 |
| 13 | Flamenco-Sandoval et al. 2007 [28] | 5755 | 7 | 1986 | 14 | with | 0 | 1 | 0.0068 |
| 14 | Freitas et al. 2010 [29] | 75 | 5 | 1962 | 19 | with | 0 | 1 | 0.024 |
| 15 | Freitas et al. 2010 [29] | 75 | 5 | 1981 | 19 | with | 1 | 0 | 0 |
| 16 | Günlü et al. 2009 [31] | 998 | 6 | 1984 | 23 | with | 0 | 0 | – |
| 17 | Hu et al. 2013 [33] | 12,104 | 5 | 1986 | 20 | with | 0 | 1 | 0.00011 |
| 18 | Jasinski et al. 2005 [34] | 900,000 | 5 | 2001 | 2 | with | 0 | 1 | 0.0024 |
| 19 | Jia et al. 2004 [35] | 312 | 8 | 1982 | 13 | with | 0 | 0 | – |
| 20 | Kane et al. 2014 [36] | 8 | 4 | 1915 | 34 | with | 1 | 0 | 0 |
| 21 | Kane et al. 2014 [36] | 8 | 4 | 1949 | 14 | with | 0 | 1 | 0.0014 |
| 22 | LaGro Jr. & DeGloria 1992 [37] | 1517 | 8 | 1968 | 17 | with | 0 | 1 | $4.9 \times 10^{-5}$ |
| 23 | Li et al. 2004 [60] | 49,286 | 7 | 1986 | 14 | with | 0 | 1 | $1.6 \times 10^{-6}$ |
| 24 | López et al. 2001 [59] | 188 | 8 | 1960 | 15 | with | 0 | 1 | 0.037 |
| 25 | López et al. 2001 [59] | 188 | 8 | 1975 | 15 | with | 0 | 1 | 0.081 |
| 26 | Mas et al. 2004 [38] | 1,932,465 | 7 | 1976 | 24 | with | 0 | 1 | 0.00034 |
| 27 | Mas et al. 2004 [38] | 1,938,326 | 7 | 1993 | 7 | with | 0 | 1 | 0.00028 |
| 28 | Matsushita et al. 2006 [39] | 2089 | 10 | 1979 | 11 | with | 0 | 1 | $6.2 \times 10^{-6}$ |
| 29 | Matsushita et al. 2006 [39] | 32 | 10 | 1990 | 6 | with | 0 | 0 | – |
| 30 | Parès-Ramos et al. 2008 [41] | 8607 | 6 | 1991 | 9 | with | 0 | 1 | 0.00096 |
| 31 | Rutherford et al. 2008 [44] | 29,613 | 5 | 1985 | 12 | with | 1 | 0 | 0 |
| 32 | Silva et al. 2011 [45] | 65 | 7 | 1990 | 13 | with | 0 | 1 | 0.016 |
| 33 | Silva et al. 2011 [45] | 122 | 7 | 1990 | 15 | with | 0 | 1 | 0.0067 |
| 34 | Silva et al. 2011 [45] | 115 | 5 | 1990 | 15 | with | 0 | 1 | 0.016 |
| 35 | Solon 2009 [46] | 379 | 6 | 1950 | 20 | with | 0 | 1 | 0.00087 |
| 36 | Solon 2009 [46] | 379 | 6 | 1970 | 20 | with | 0 | 1 | $5.7 \times 10^{-5}$ |
| 37 | Takada et al. 2010 [12] | 100 | 5 | 1947 | 15 | with | 0 | 1 | 0.0028 |
| 38 | Takada et al. 2010 [12] | 100 | 5 | 1962 | 13 | with | 0 | 1 | 0.011 |
| 39 | Takada et al. 2010 [12] | 100 | 5 | 1975 | 22 | with | 0 | 1 | 0.024 |
| 40 | Tang et al. 2007 [61] | 2017 | 7 | 1979 | 11 | with | 0 | 1 | 0.012 |
| 41 | Tang et al. 2007 [61] | 2017 | 7 | 1990 | 10 | with | 0 | 1 | 0.025 |
| 42 | Weng 2001 [48] | 15,112 | 7 | 1989 | 8 | with | 0 | 1 | 0.012 |
| 43 | Yu & Ng 2006 [49] | 1231 | 6 | 1988 | 5 | with | 0 | 0 | – |
| 44 | Yu & Ng 2006 [49] | 1231 | 6 | 1993 | 5 | with | 0 | 1 | 0.048 |
| 45 | Yu & Ng 2006 [49] | 1231 | 6 | 1998 | 4 | with | 0 | 0 | – |
| 46 | Yuechen 2008 [50] | 5,308,690 | 6 | 1989 | 10 | with | 0 | 1 | 0.0033 |
| 47 | Yuechen 2008 [50] | 5,308,690 | 6 | 1999 | 4 | with | 0 | 1 | 0.0050 |
| 48 | Zarin et al. 2001 [51] | 523 | 5 | 1976 | 15 | with | 0 | 1 | 0.0017 |
| 49 | Barima et al. 2010 [22] | ? | 4 | 1986 | 16 | without | 0 | 1 | * |
| 50 | Barima et al. 2010 [22] | ? | 4 | 1986 | 16 | without | 0 | 0 | * |
| 51 | Guan et al. 2011 [30] | 431 | 6 | 1976 | 11 | without | 0 | 1 | * |
| 52 | Guan et al. 2011 [30] | 431 | 6 | 1987 | 10 | without | 0 | 1 | * |
| 53 | Guan et al. 2011 [30] | 431 | 6 | 1997 | 9 | without | 0 | 1 | * |
| 54 | Hall et al. 1991 [32] | 409 | 6 | 1973 | 10 | without | 0 | 1 | * |
| 55 | Hall et al. 1991 [32] | 534 | 6 | 1973 | 10 | without | 0 | 1 | * |
| 56 | Mendoza et al. 2011 [40] | 4000 | 9 | 1975 | 11 | without | 0 | 1 | * |
| 57 | Mendoza et al. 2011 [40] | 4000 | 9 | 1986 | 10 | without | 0 | 1 | * |
| 58 | Mendoza et al. 2011 [40] | 4000 | 9 | 1996 | 4 | without | 0 | 1 | * |
| 59 | Mendoza et al. 2011 [40] | 4000 | 9 | 2000 | 3 | without | 0 | 1 | * |
| 60 | Peña et al. 2007 [42] | 641 | 7 | 1956 | 44 | without | 0 | 1 | * |
| 61 | Pueyo & Alados 2007 [43] | 457 | 5 | 1957 | 41 | without | 1 | 0 | * |
| 62 | Thomlinson et al. 1996 [47] | 43 | 7 | 1936 | 52 | without | 0 | 0 | * |

Note *: Error estimation was not conducted because area data were not available for the matrix.

**Table 3.** Probability(%) of obtaining a positive and small-negative solution with random matrices. Former: positive solution; latter: small-negative solution. Numbers with one decimal place indicate results with 1000 trials and the others indicate results with 100 trials.

| Matrix Size | Power Root | | | | | | | |
|---|---|---|---|---|---|---|---|---|
| | 3 | 4 | 5 | 7 | 10 | 13 | 20 | 30 |
| 2 | 70.9, 13.0 | 59.1, 5.5 | 63.9, 12.7 | 60.0, 12.7 | 53.3, 6.2 | 56.0, 12.3 | 51.9, 4.7 | 51.4, 4.6 |
| 3 | 8.4, 24.0 | 6.6, 12.9 | 6.1, 17.5 | 5.2, 17.3 | 4.9, 16.5 | 4.7, 22.6 | 4.3, 26.9 | 4.3, 35.7 |
| 4 | 0.2, 7.6 | 0.0, 4.6 | 0.0, 5.4 | 0.0, 6.6 | 0.0, 9.4 | 0.0, 13.5 | 0.0, 21.1 | 0.0, 32.7 |
| 5 | 0.0, 1.6 | 0.0, 1.2 | 0.0, 1.4 | 0.0, 1.8 | 0.0, 4.5 | 0.0, 8.4 | 0.0, 15.8 | 0.0, 28.4 |
| 6 | 0.0, 0.1 | 0.0, 0.0 | 0.0, 0.0 | 0.0, 0.2 | 0.0, 1.2 | 0.0, 2.9 | 0.0, 10.2 | 0, 17 |
| 7 | 0.0, 0.0 | 0.0, 0.0 | 0.0, 0.0 | 0.0, 0.0 | 0.0, 0.9 | 0.0, 2.5 | 0, 6 | 0,14 |
| 8 | 0.0, 0.0 | 0.0, 0.0 | 0.0, 0.0 | 0.0, 0.0 | 0.0, 0.3 | 0,1 | – | – |
| 9 | 0.0, 0.0 | 0.0, 0.0 | 0.0, 0.0 | 0.0, 0.0 | 0.0, 0.0 | 0,0 | – | – |

In contrast, the probability of obtaining a positive solution with a modified random matrix was generally higher than that with a random matrix (Table 4).

**Table 4.** Probability(%) of obtaining a positive and small-negative solution with modified random matrices. Former: positive solution; latter: small-negative solution. Numbers with one decimal place indicate results with 1000 trials and the others indicate results with 100 trials.

| Matrix Size | Power Root | | | | | | | |
|---|---|---|---|---|---|---|---|---|
| | 3 | 4 | 5 | 7 | 10 | 13 | 20 | 30 |
| 2 | 100.0, 0.0 | 100.0, 8.7 | 100.0, 0.0 | 100.0, 0.0 | 100.0, 8.0 | 100.0, 0.0 | 100.0, 8.9 | 100.0, 9.5 |
| 3 | 38.8, 61.1 | 35.5, 66.3 | 33.7, 66.0 | 32.7, 67.0 | 31.7, 69.0 | 31.0, 68,7 | 30.2, 70.4 | 29.7, 70.9 |
| 4 | 3.6, 94.4 | 2.8, 95.9 | 2.6, 96.4 | 2.3, 96.8 | 2.0, 97.3 | 1.8, 97.5 | 1.8, 97.5 | 1.7, 97.6 |
| 5 | 0.0, 93.1 | 0.0, 95.0 | 0.0, 96.2 | 0.0, 97.4 | 0.0, 98.2 | 0.0, 98.3 | 0.0, 98.4 | 0.0, 98.4 |
| 6 | 0.0, 90.4 | 0.0, 92.8 | 0.0, 94.4 | 0.0, 96.0 | 0.0, 97.4 | 0.0, 97.5 | 0.0, 97.6 | 0, 99 |
| 7 | 0.0, 86.3 | 0.0, 89.7 | 0.0, 91.1 | 0.0, 93.8 | 0.0, 96.0 | 0.0, 96.3 | 0, 96 | 0, 97 |
| 8 | 0.0, 78.4 | 0.0, 83.2 | 0.0, 86.7 | 0.0, 90.9 | 0.0, 92.9 | 0, 94 | – | – |
| 9 | 0.0, 75.1 | 0.0, 79.1 | 0.0, 83.7 | 0.0, 87.8 | 0.0, 90.8 | 0, 93 | – | – |

For a matrix size of 2, the probability was 100%. Similarly, the probability of obtaining a small-negative solution with a modified random matrix was higher than that with a random matrix, and exceeded 90% for matrix sizes greater than 4 (Table 4). However, the probability was zero for a matrix size of 2 and the power root was odd.

## 4. Discussion

The possibility of obtaining yearly transition probability matrices with real field data set, random matrices, and modified random matrices using TAM, was high (54 in 62 matrices; 87%, Table 2), low (Table 3), and relatively high (Table 4), respectively. These results suggest that TAM may provide suitable solutions using transition probability matrices with relatively large diagonal elements, which is common in LUCC studies.

The low probability of obtaining a positive or small-negative solution from a random matrix, especially for a matrix greater than 5 × 5 (Table 3), suggests that a yearly transition matrix with real and positive elements cannot always be derived from a transition matrix. However, the probability of obtaining a positive or small-negative solution from a modified random matrix (relatively large diagonal elements) exceeded 90% in most cases (Table 4). The diagonal elements of the transition probability matrix tend to be relatively large in land-use dynamics analysis because land-use patterns are fairly constant over a short period. Consequently, TAM should derive yearly transition matrices from the transition probability matrices for land-use analysis.

In cases where the target location experiences drastic changes and the self-replacement rate is low, the diagonal elements of the transition probability matrices tend to be not relatively large. In the empirical verification, annual transition probability matrices were not obtained from No. 9, 16, 19, 29, 43, 45, 50, and 62 matrices in Table 2. These matrices were from studies on urbanization (No. 43 &45; [49]), rapid agricultural or industrial land-use change (No. 9; [27], No. 19; [35],

No. 29; [39], No. 50; [22]) and long-term studies (No. 16; [31], No. 62; [47]). For these types of studies, it is possible that TAM cannot obtain an applicable solution of yearly transition matrix.

The estimated error with area projection between real and annualized transition probability matrices was lower than 0.05 in most cases (Table 2). The estimated error did not correlate with the study area size and the number of classes, but it was negatively correlated with the study period. TAM includes errors attributable to calibration, in which negative elements close to zero are treated as zero [12]. Nevertheless, these results indicated that the calibration error is small and independent of study area size and number of classes.

We could not calculate the annual transition probability matrix from several large (containing many classes) and long-term matrices, such as [55,56,58,62], because their estimated time of calculation exceeded a month. For example, even with the newest PC (AMD Ryzen5 3600x, 3.8GHz) configuration, the calculation of an annual transition probability matrix from the transition probability matrix from Ojeda-Revah et al. [62], whose matrix size (number of categories) was 10 and study period was 24 years, was estimated to consume more than 1000 days. Please note that TAM will check all the possible $m^n$ solutions for $n \times n$ transition probability matrix whose duration is $m$ years. Therefore, other algorithms will be needed to speed up the calculation for large and long-term transition probability matrices.

## 5. Conclusions

In this study, empirical verification based on transition matrices from previous land-use studies and Monte-Carlo simulations were conducted to estimate the probability of obtaining an appropriate yearly transition probability matrix with TAM. This study has revealed that (1) the possibility of obtaining yearly transition probability matrices with real field data set is high, (2) the theoretical possibility of obtaining yearly transition probability matrices is low, as shown in Monte-Carlo simulation with random matrices, and (3) the difference between real and theoretical results may be explained by high possibility of obtaining yearly transition probability matrices in Biased Monte-Carlo simulation, suggesting that the possibility of obtaining yearly transition probability matrices is high when the diagonal elements of the transition probability matrix were relatively larger than non-diagonal elements. The diagonal elements of the transition probability matrix tend to be relatively large in most cases in LUCC studies. However, the diagonal elements of the transition probability matrices tend to be not relatively large in cases where the target location experiences drastic changes such as urbanization, rapid agricultural or industrial land-use change and long-term research. For these types of studies, TAM may not be able to obtain an applicable solution of yearly transition matrix.

This study suggests that TAM is applicable for many transition probability matrices and may contribute to land-use dynamics analysis as a powerful tool.

**Author Contributions:** Conceptualization, S.F.H. and T.T.; methodology, S.F.H. and T.T.; formal analysis, T.T.; investigation, S.F.H.; writing—original draft preparation, S.F.H.; writing—review and editing, T.T.

**Funding:** This research was funded in part by Grants-in-Aid from D-04 of the Research Institute for Humanity and Nature and by JSPS KAKENHI Grant Number 22570011.

**Acknowledgments:** We express our sincerest gratitude to Jan Bogaert, Toshihiko Hara, Norio Yamamura and Takashi S. Kohyama for their helpful suggestions.

**Conflicts of Interest:** The authors declare no conflict of interest.

## Abbreviations

The following abbreviations are used in this manuscript:

LUCC    Land-Use/Cover Change
TAM      Takeda et al. (2010)'s Annualization Method

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
