# Peer review of "Probability of Deriving a Yearly Transition Probability Matrix for Land-Use Dynamics"

_sustainability, doi:10.3390/su11226355_

Round 1

Reviewer 1 Report

This is a very interesting research you conducted and very useful in developing further works in a subject of LULC changes. I don't have many remarks, either I have found any mistakes, it is rather a request for some clarifications. 

[49] - I don't understand what exactly the imaginary elements are

[96] - you mention a software, if your methodology is going to have followers it would be useful to mention the name and maybe a possibility to use any others

Table 2 - is a small negative solution "0" and positive is "1"? This is not clear

Table 2 - if you don't know the size of the error how can you compare the results with the ones the error is known? 

Table 2 - [156] - where is, in this table, the 87% probability?  for me it is not clear

[173] - No9 & 29 matrices - the error is unknown, I don't think you should discuss them as a notable result

[192-193] - it would be a very valuable remark if you tell more about limitations, what data can be analysed which can't, it is very important for the followers 

Author Response

We are grateful for your knowledgeable comments and are especially indebted to your kind patience and academic rigidity. Revisions and corrections have been made in accordance with your helpful suggestions as follows.

Comment 1. [49] - I don't understand what exactly the imaginary elements are.

Response 1. We have revised the manuscript as “matrices with imaginary elements or negative elements”→”matrices with complex numbers or negative numbers” in line 49.

Comment 2. [96] - you mention a software, if your methodology is going to have followers it would be useful to mention the name and maybe a possibility to use any others.

Response 2. We have added the name of software and available URL in lines 66-67.

Comment 3. Table 2 - is a small negative solution "0" and positive is "1"? This is not clear.

Response 3. Table 2 was misleading because the number of positive and small negative solutions are mixed. We have separated the row of positive and small negative solutions in Table 2.

Comment 4. Table 2 - if you don't know the size of the error how can you compare the results with the ones the error is known?

Response 4. We can estimate the errors only when we could obtain at least one positive or small negative annual transition matrix and area data. It is impossible to compare the error if we could not obtain annual transition matrix.

Comment 5. Table 2 - [156] - where is, in this table, the 87% probability? for me it is not clear.

Response 5. The 87% probability was calculated as the number of probability transition matrices that was obtained a positive or small-negative solution (54) over the number of all matrices used in empirical verification (62), as shown in line 130. We have revised the manuscript as “54 in 62 matrices; 87%” in line 157.

Comment 6. [173] - No9 & 29 matrices - the error is unknown, I don't think you should discuss them as a notable result

Response 6. We have deleted the lines 173-175.

Comment 7. [192-193] - it would be a very valuable remark if you tell more about limitations, what data can be analysed which can't, it is very important for the followers

Response 7. We have moved these lines to “Conclusions” section and revised according to the reviewer's comment.

Reviewer 2 Report

The article is very interesting. Detailed evaluation of mathematical methods available unambiguous results.

In this study, empirical verification based on transition matrices from previous land-use studies and Monte-Carlo simulations were conducted to estimate the probability of obtaining an appropriate yearly transition probability matrix.

The practical appeal is the absence of conclusions. Please add to publications.

Author Response

We are grateful for your knowledgeable comments and are especially indebted to your kind patience and academic rigidity. Revisions and corrections have been made in accordance with your helpful and enlightening suggestions as follows.

Comment 1. The practical appeal is the absence of conclusions. Please add to publications.

Response 1. We have added “Conclusions” section.